# ClO_2_-Mediated Oxidation of the TEMPO Radical: Fundamental Considerations of the Catalytic System for the Oxidation of Cellulose Fibers

**DOI:** 10.3390/molecules28186631

**Published:** 2023-09-15

**Authors:** Laura Giraldo Isaza, Gérard Mortha, Nathalie Marlin, Florian Molton, Carole Duboc

**Affiliations:** 1Institute of Engineering, University Grenoble Alpes, CNRS, Grenoble INP, LGP2, F-38000 Grenoble, France; 2Department of Molecular Chemistry, University Grenoble Alpes, CNRS, DCM, F-38000 Grenoble, France

**Keywords:** TEMPO oxidation, chlorine dioxide, EPR spectroscopy, UV–Vis spectroscopy

## Abstract

The reaction mechanism of ClO${}_{2}$-mediated TEMPO oxidation was investigated by EPR spectroscopy and UV–Vis spectroscopy in the context of an alternative TEMPO sequence for cellulose fiber oxidation. Without the presence of a cellulosic substrate, a reversibility between TEMPO and its oxidation product, TEMPO${}^{+}$, was displayed, with an effect of the pH and reagent molar ratios. The involvement of HOCl and Cl${}^{-}$, formed as byproducts in the oxidation mechanism, was also evidenced. Trapping HOCl partly inhibits the reaction, whereas adding methylglucoside, a cellulose model compound, inhibits the reversibility of the reaction to TEMPO.

## 1. Introduction

Since first proposed as a selective pathway for the oxidation of primary alcohol groups in polysaccharides [1], the stable aminoxyl TEMPO radical has advanced as a recourse for the catalytic formation of C6 carboxylate groups in cellulose [2]. For this system, the nitroxyl radical is activated by NaClO, resulting in an oxoammonium salt (TEMPO${}^{+}$) which then oxidizes the primary hydroxyl groups in the D-glucose units into negatively charged carboxyl entities through C6 aldehydes [3]. Finally, the reduced TEMPO is oxidized by NaBrO, a byproduct from the oxidation of NaBr by NaClO, to recover TEMPO${}^{+}$ [4] (Figure 1).

Related to a notable regioselectivity [5,6] paired with a high reaction rate and yield at alkaline conditions (pH 10) [6,7], a conventional TEMPO-mediated oxidation supports the production of cellulose fibers with a marked carboxyl and aldehyde content [8]. Thus, this treatment generates highly oxidized fibers, with good dispersibility [5], fibrous morphology, preserved crystallinity [8], and limited depolymerization [6]. Hence, TEMPO oxidation has been consolidated as a widely researched proposal for the chemical modification of cellulose, in particular as a preparation for mechanical fiber disruption [9,10,11,12,13,14].

Specially, anionic carboxyl groups settled in the fibers frame through TEMPO oxidation enhance their overall negative charge, generating an electrostatic repulsion that enlarges the interfibrillar region, signifying a less energy-demanding disintegration [9]. After applying high-shearing forces within the fiber section, microfibrillated cellulose (MFC) is obtained [15]. This derivative synthesizes cellulose properties with technical characteristics, such as high crystallinity, high specific surface area, biocompatibility, low density, and good tensile strength [16], enlarging the perspective for the development of novel bio-based materials.

However, from an economical and environmental viewpoint, a bromide-free cycle is more viable, since NaBr leads to the production of toxic halogenated organic compounds in the waste stream [17], also related to machinery corrosion issues. This mentioned drawback, in combination with a costly expense for the TEMPO catalyst and an elevated energy use during fiber microsizing [9], restricts future industrial scale advances in the sustainable production of biodegradable microfibrillated cellulose composites.

Proposed to replace NaBr, chlorine dioxide (ClO${}_{2}$) appears as a powerful oxidant [18], which reacts instantly with the nitroxyl compound [19]. Its remarkable potential relies on a limited reactivity against cellulosic fibers, limiting their depolymerization [20]. Moreover, ClO${}_{2}$ is a widely used bleaching agent, opening the possibility to integrate this oxidation cycle in current paper mill facilities to improve the valorization of cellulose fibers.

The total oxidation of TEMPO into its oxoammonium form can be promoted by an equimolar amount of ClO${}_{2}$ due to a linear dependence in the compound absorption [19]. In the reaction, a charge–transfer complex is established between TEMPO and ClO${}_{2}$, which promotes the conversion of TEMPO via inner-sphere one-electron oxidation through an intermediate [19], as illustrated in Figure 2. Notably, the radical reacts rapidly with ClO${}_{2}$ at room temperature, indicated by an instant dissipation of the yellowish ClO${}_{2}$ coloration [21]. The reaction proceeds with a pseudo-first-order rate constant of 3.7 × 10${}^{5}$ s${}^{-1}$ ((TEMPO): 0.01 M) [19], regulated by the TEMPO concentration, temperature, and pH (pH 8) [22].

After a separated activation stage, the oxoammonium cation can act as a direct oxidant for the C${}_{6}$ alcohol conversion in the target cellulose molecule [20,21,22]. Owing to a specific TEMPO conversion by ClO${}_{2}$, cellulose oxidation is improved remarkably [22], ensuring a high percentage of carboxyl conversion [20]. For instance, Pääkkönen et al. [21] produced cellulose fibers with a conversion degree of 1.03 mmol COOH/g pulp, while Tienvieri et al. [22] ensured a carboxyl content from 0.69–0.85 mmol COOH/g pulp, and Dollie et al. [23] achieved an oxidation yield of 0.6 mmol COOH/g pulp starting from unbleached fibers and semibleached fibers (with residual lignin). In this study, ClO${}_{2}$ was applied as a synergistic TEMPO activator and bleaching compound in a complementary delignification and oxidation system to diversify the processing stages in paper mills.

Although a ClO${}_{2}$-TEMPO oxidation process conveys an innovative pathway for a high-scale performable microfibrillated cellulose production, a systematic understanding of this recent cellulose oxidation sequence has not been further studied. For instance, an integral kinetic study of the chemical system is required to better understand the interaction between ClO${}_{2}$ and TEMPO, define the best pH and molar ratio values, evaluate the influence of ClO${}_{2}$ degradation compounds during the TEMPO redox cycle, observe the general reaction dynamics, and assess the influence of noncellulosic compounds potentially present, such as lignin. Thus, the oxidation system will be studied first without any substrate and then with methyl glucoside (cellulose model compound) and vanillin (lignin model compound).

## 2. Results and Discussion

### 2.1. Investigation of the ClO_2_-Based TEMPO Activation through EPR Spectroscopy

#### 2.1.1. Effect of pH and TEMPO/ClO${}_{2}$ Molar Ratio

In this section, the ClO${}_{2}$-based TEMPO activation is analyzed in the absence of cellulose or lignin model compounds. EPR signals for the nitroxyl radical, prior and after oxidation at pH 8 and 10 with an equimolar amount of ClO${}_{2}$, are shown in Figure 3. After ClO${}_{2}$ is added, the TEMPO signal intensity is diminished, indicating the formation of a chemical complex (derivation of TEMPO${}^{+}$ not visible by EPR), regardless of the reaction pH value.

By qualitatively comparing the spectral lines, oxidation at pH 8 proves a lower signal intensity, meaning a more complete conversion of TEMPO. This observation is validated with the quantification of the spin concentration, in which the presence of the radical was reduced in 83.4% when the reaction occurred at pH 8, contrasting with a depletion of 72.1% after TEMPO activation at pH 10. In this sense, the conversion yield is higher at pH 8, even if the oxidative potential of ClO${}_{2}$ is not pH-dependent [24].

This supports the assertion that a bromide-free nitroxyl oxidation should proceed at almost neutral pH for a more optimal activation [17], specifically for a ClO${}_{2}$-based TEMPO oxidation. This could be linked to production of hypochlorous acid (HOCl) as a byproduct from the alkaline oxidation of ClO${}_{2}$ [25]. Indeed, HOCl (and not ClO${}^{-}$) is recognized as a direct oxidant for the aminoxyl compound [26]. Regarding its pKa (7.6), at a more alkaline pH level, HOCl starts to dissociate, meaning that around pH 8, there is still a relevant presence of HOCl, which can act as a complementary activation agent during ClO${}_{2}$-mediated TEMPO oxidation. Thus, a minor activation at pH 10 could be related to a lower HOCl concentration, which limits a possible parallel oxidation reaction.

As shown in Figure 4, a receding TEMPO signal is identified with a greater TEMPO/ClO${}_{2}$ ratio, with a reduction of 88.87% of its initial concentration, meaning a better conversion at a higher oxidant proportion. Significantly, TEMPO signal intensity proves a proportional relation between the oxidant amount and nitrosonium converted; however, the effect of doubling the oxidant ratio does not greatly improve the reaction yield (83.4% reduction at TEMPO/ClO${}_{2}$ 1:1). Despite this, a defined ClO${}_{2}$ presence is required, as the ClO${}_{2}$ active concentration is impacted by its alkaline disproportionation into ClO${}_{2}$${}^{-}$ and ClO${}_{3}$${}^{-}$ [27].

#### 2.1.2. TEMPO Regeneration during ClO${}_{2}$ Activation

The signal sweep period was extended to evaluate the stability of the system. The nitroxyl EPR spectra, taken instantly, then at 3 min and 5 min, are displayed in Figure 5 for the 1:1 and 1:2 TEMPO/ClO${}_{2}$ reactions. An increase in the nitroxyl signal can be noted after its initial instant decay (formation of TEMPO${}^{+}$). After about 5 min, the recovery of TEMPO is important, especially with a TEMPO/ClO${}_{2}$ ratio of 1:2. Therefore, it seems that a higher ClO${}_{2}$ concentration affects TEMPO recovery, as a more progressive augmentation of its EPR signal is observed when TEMPO has been activated by a higher mount of ClO${}_{2}$.

In detail, for the TEMPO/ClO${}_{2}$ 1:1 reaction, the first measurement displays 16.6% of the initial TEMPO concentration, and after 3 min, 29.1% of the radical is detected, and in 5 min, approximately 60.1% of the nitroxyl signal reappears. Conversely, with a higher ClO${}_{2}$ ratio, 11.1% of the signal is captured in the first instance; then, it increases at 17.7% after 3 min, until reaching a TEMPO recovery of 80.4%.

ClO${}_{2}$-mediated TEMPO activation proves a chemical reversibility behavior adjusted to a redox displacement [28], as seen in Figure 6. Without a cellulosic substrate, restitution of TEMPO is attained by the reduction in TEMPO${}^{+}$. Considering the ClO${}_{2}$ disproportionation in aqueous medium [27] and the decomposition of ClO${}_{2}$${}^{-}$ in alkaline medium [29], the generated chloride (Cl${}^{-}$) from these reactions (and initial Cl${}^{-}$ present in the ClO${}_{2}$ solution) appears as a potential reducer of TEMPO${}^{+}$, as shown in the following chemical pathway:$$\mathrm{TEMPO}+\mathrm{ClO}{}_{2}⟶\mathrm{TEMPO}{}^{+}+\mathrm{ClO}{}_{2}^{-}\;Oxidation\;of\;TEMPO$$
$$\mathrm{ClO}{}_{2}^{-}⟶\mathrm{Cl}{}^{-}+O{}_{2}\;Chloride\;formation\;from\;chlorite$$
$$2\mathrm{ClO}{}_{2}+H{}_{2}O⟶\mathrm{ClO}{}_{3}^{-}+\mathrm{ClO}{}_{2}^{-}+2H{}^{+}\;Disproportionation\;of\;ClO{}_{2}$$
$$3\mathrm{ClO}{}_{2}^{-}⟶\mathrm{Cl}{}^{-}+2\mathrm{ClO}{}_{3}^{-}\;Decomposition\;of\;chlorite\;in\;alkaline\;medium$$
$$\mathrm{TEMPO}{}^{+}+\mathrm{Cl}{}^{-}⟶\mathrm{TEMPO}+\frac{1}{2}\mathrm{Cl}{}_{2}\;Regeneration\;of\;TEMPO\;from\;TEMPO{}^{+}$$

Alternatively, the nitroxyl radical can be recovered by the following chemical routes. Respectively, the main mechanism is the comproportionation reaction between the oxoammonium cation (TEMPO${}^{+}$) and the hydroxylamine (TEMPO-OH) derived from the cyclic nitroxide after its oxidation. From the equations, it can be deduced that the recovery of TEMPO is promoted by a higher presence of chloride. Thus, with a surplus of ClO${}_{2}$ during TEMPO activation, the reaction is favored towards the generation of TEMPO${}^{+}$ and chloride. Then, the chemical equilibrium of the system is shifted to an upgraded TEMPO recovery.
$$2\mathrm{Cl}{}^{-}+\mathrm{TEMPO}{}^{+}⟶\mathrm{TEMPO}-\mathrm{OH}+\mathrm{Cl}{}_{2}\;Alternative\;pathway\;for\;TEMPO{}^{+}\;reduction$$
$$\mathrm{TEMPO}{}^{+}+\mathrm{TEMPO}-\mathrm{OH}⟶2\mathrm{TEMPO}+H{}^{+}\;Comproportionation\;reaction$$

To confirm this hypothesis, the Cl${}^{-}$ generated is precipitated by silver nitrate (AgNO${}_{3}$). The TEMPO signal recovery, after oxidation, in the absence of Cl${}^{-}$ is presented in Figure 7. The spin density does not increase with time, only achieving 16.3% of TEMPO recovery after 5 min. This small TEMPO recovery could be related to an insufficient Cl${}^{-}$ precipitation. It is then demonstrated that the suppression of the reducer limits the one e${}^{-}$ transfer, which promotes the reconversion of TEMPO${}^{+}$ back to its nitroxyl state. Although the EPR signal indicates the limited reappearance of TEMPO under the inhibition of chloride, this reaction route is not clear. It has been reported that the reaction between the oxoammonium salt and chloride is a double bond addition in which TEMPO${}^{+}$ acts as the electrophile and chloride as the nucleophile [30]. In both reactions, gases are generated (O${}_{2}$ and Cl${}_{2}$), but the close conditions of the EPR acquisition do not allow to observe the formation of bubbles.

**Figure 6 molecules-28-06631-f006:**
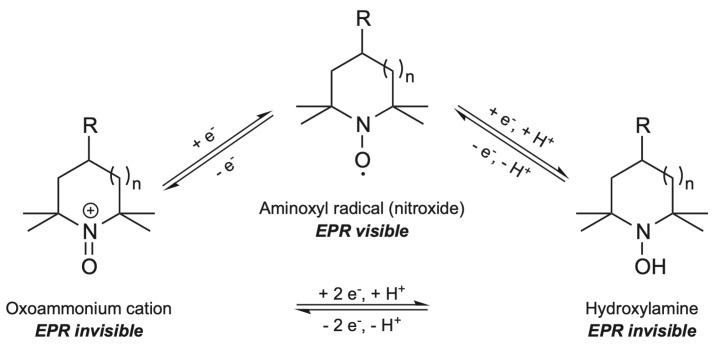
EPR detection of redox species associated with aminoxyl radicals, taken from [31].

#### 2.1.3. Assessment of a Cellulosic Model Compound during ClO_2_-Based TEMPO Oxidation

In the presence of cellulose, TEMPO-OH is generated from the reaction of TEMPO${}^{+}$ with the C6 OH group present in the cellulosic anhydroglucose unit. It has been reported in the literature that in acidic medium, TEMPO might be formed from TEMPO-OH by direct oxidation of the latter; and that in alkaline medium, TEMPO might be generated by comproportionation involving a reaction between TEMPO${}^{+}$ and TEMPO-OH. Specifically, the disproportionation of TEMPO, resulting in the reaction of TEMPO${}^{+}$ and TEMPO-OH, is enhanced at acidic pH [17]. The comproportionation reaction implies the inverse route [32].

Ergo, a cellulose model compound (methyl-α-glucoside) was included as a substrate to promote the TEMPO${}^{+}$ reaction and its reduction into TEMPO-OH in parallel to simulate a TEMPO response in this condition. By excluding ClO${}_{2}$ from the solution (only TEMPO and methyl-α-glucoside were mixed together), it was observed that the nitroxyl signal of TEMPO did not present any changes. This can be explained by the absence of TEMPO${}^{+}$ regarding the lack of an oxidizing agent to react with TEMPO and therefore the nonpresence of TEMPO-OH as a reaction product from TEMPO${}^{+}$ reduction. When adding ClO${}_{2}$ to this system, the EPR spectra after TEMPO-ClO${}_{2}$ and the cellulose model compound reaction, precised in Figure 8, proves an important decay of the TEMPO signal and no noticeable TEMPO restoration. This can be attributed to (1) the absence of TEMPO${}^{+}$, since it fully reacts with the methylglucoside to generate TEMPO-OH, and (2) the lack of a TEMPO-OH reoxidation into TEMPO${}^{+}$ or TEMPO, involving NaClO or another oxidizing species [33].

Since TEMPO-OH is the final reaction product in the present reaction, it is evident that the comproportionation reaction is not a prevailing pathway for the restitution of TEMPO, which would compete with the reaction of TEMPO${}^{+}$ with the methylglucoside, forming TEMPO-OH. In fact, TEMPO restoration by a comproportionation reaction is favored by strong alkaline conditions [32]. Thus, at a pH of 8 (chosen in this study as the optimal range for ClO${}_{2}$-based TEMPO oxidation of cellulose, as it proves a better yield at low alkaline pH), a comproportionation route is not favored in the proposed catalytic sequence. Moreover, other studies seem to confirm that such reaction is too slow to greatly affect the TEMPO restoration rate [34]. Consequently, within the context of cellulose oxidation, the addition of a primary oxidant (sodium hypochlorite) [33] or a copper(II) salt [35] to reoxidize TEMPO-OH back to TEMPO to continue with cellulose modification is recognized.

#### 2.1.4. Assessment of a Lignin Model Compound during ClO${}_{2}$-Based TEMPO Oxidation

Lignin in unbleached kraft pulps can be distributed along the fibrils surface, affecting their chemical modification. Lignin is structured by phenylpropanoid units, with reactive functional groups prone to react with TEMPO and oxidizing agents during lignocellulosic oxidation [36]. More precisely, lignin can be oxidized and depolymerized into water-soluble lignin-derived compounds during TEMPO oxidation, without radical changes in the lignin phenolic structure [37]. Since nitroxyl activation involves the catalytic action of ClO${}_{2}$, an important delignification agent [38], the degradation of the phenolic frame can prevail as a parallel reaction. With respect to the disposition of ClO${}_{2}$ to oxidize TEMPO and lignin, a competition might be seen. Hence, ClO${}_{2}$ selectivity is assessed to define the main reaction route during the modified ClO${}_{2}$-based TEMPO cellulose oxidation sequence.

Accordingly, the nitroxyl conversion was conducted in the presence of a phenolic lignin model compound: vanillin (4-hydroxy-3-methoxybenzaldehyde). The test was carried out with a 2:1 molar ratio regarding ClO${}_{2}$ [39], since ClO${}_{2}$ oxidation in acidic medium of a free phenolic unit involves two ClO${}_{2}$ molecules per aromatic unit, generating one chlorite anion (ClO${}_{2}$${}^{-}$) and one hypochlorite anion (ClO${}^{-}$) as main reaction products [39]. At first, the EPR signal for the ClO${}_{2}$-vanillin reaction is seen in Figure 9. The red spectral line corresponds to the ClO${}_{2}$ radical EPR pattern [40]. During vanillin oxidation, ClO${}_{2}$ is totally consumed; therefore, a quantitatively small EPR signal for ClO${}_{2}$ is observed (black line) with a reduction of 92.7% in the spin concentration, implying a compelling chemical interaction.

Then, the spectral line for the TEMPO-ClO${}_{2}$ system was studied in the presence of an equimolar amount of vanilline, as shown in Figure 10. Minimum EPR spectral intensity difference is appreciated in contrast with the nitroxyl spectrum, with a spin concentration reduction of only 89% initial TEMPO, meaning a low formation of TEMPO${}^{+}$. Forthwith, it can be deduced that ClO${}_{2}$ reacts faster with vanilline than with TEMPO. Indeed, it is known that the reaction between ClO${}_{2}$ and free phenolic groups in lignin is very fast [41] due to their strong affinity. Since ClO${}_{2}$ was not introduced in excess of the phenolic compound, the amount of free oxidant was immediately depleted, leaving a lower quantity to react with TEMPO, approximately 7.3% of the initial ClO${}_{2}$ spin concentration. This observation suggests that within a ClO${}_{2}$-based transformation of unbleached kraft fibers, there would be a preferential lignin oxidation route.

#### 2.1.5. Influence of HOCl as a Co-Oxidant

Despite the attack by ClO${}_{2}$ of organic substrates bearing labile protons (like phenolic groups), this reaction consists in a proton abstraction and a one-electron transfer with the substrate, releasing one chlorite ion and an organic R^*o*^ radical, and then the further reaction of R^*o*^ with a second molecule of ClO${}_{2}$ releases HOCl.

As byproduct, HOCl represents a selective oxidant that can also react with TEMPO to form an oxoammonium salt [26], which was previously validated by Pääkkönen et al. [26]. Considering (1) the possible involvement of HOCl in the ClO${}_{2}$/TEMPO oxidation system, (2) the role of HOCl in cellulose oxidation involving the redox couples (TEMPO${}^{+}$/TEMPO-OH) and (HOCl/Cl${}^{-}$), and (3) the possible synergies between all the different oxidants in presence, it was interesting to study the effect of the presence or absence of HOCl in the catalytic system. To this end, the reaction of TEMPO with ClO${}_{2}$ was probed with a HOCl masking agent, dimethyl sulfoxide (DMSO). The results are displayed in Figure 11.

From the variance in the intensity of the resonance spectrum, a comparable pattern to that without DMSO is observed. Certainly, the decline in the EPR signal intensity is lower when HOCl is trapped: 70.3% of initial TEMPO concentration is achieved. This is also noted when HOCl is added alone (without ClO${}_{2}$${}^{-}$; see Figure 11), completing the oxidation of 80.5% of initial TEMPO. As proven, TEMPO concentration in the medium is marginally lower when HOCl contributes to the oxidation cycle. Thus, in its absence, the results are similar to the EPR lines after a 1:1 TEMPO/ClO${}_{2}$ reaction. These features support the influence of HOCl as a correlative activator factor, distinctly at a pH closer to 7, still presenting a relatively high concentration at mild alkaline conditions (14%) [26].

### 2.2. UV–Visible Spectroscopy (UV–Vis)

ClO${}_{2}$ exhibits a large absorption peak centered around 360 nm, not superimposed with the TEMPO peak centered at 245 nm. Accordingly, the reactants and TEMPO${}^{+}$ spectral lines resulting from TEMPO-ClO${}_{2}$ reactions are shown in Figure 12. The main scope of this analysis was to show the ClO${}_{2}$ remaining concentration after TEMPO oxidation, as its signal intensity is smaller than the radical; hence, it was not observable in the managed EPR spectrum intensity scale. As the ClO${}_{2}$ curve gradually declines, it proves that an equimolar ratio activates the radical [19]. When an excess of ClO${}_{2}$ is used (1:2 TEMPO/ClO${}_{2}$), it seems that the reaction is further completed and a surplus of ClO${}_{2}$ is left, possibly available for other reaction. Therefore, a higher ClO${}_{2}$ concentration can further enhance the conversion. In both proportions, a pronounced slope at 290 nm and transition at 245 nm are displayed, stating the formation of the nitrosonium ion.

## 3. Materials and Methods

### 3.1. Electron Paramagnetic Resonance (EPR) Principle for TEMPO Oxidation Detection

The EPR technique uses microwave radiation to probe species such as radicals in the presence of an externally static applied magnetic field. EPR is conditioned to the electron spin (e${}^{-}$) and its magnetic moment (μ${}_{e}$). When an e${}^{-}$ is placed inside a magnetic field, the Zeeman effect induces two spin states of opposite energy. The lower energy state occurs when the electron’s magnetic moment (μ) is aligned with the magnetic field, and a higher energy state is reflected when μ is not oriented in parallel.

For a molecule with unpaired e${}^{-}$ inside a magnetic field, an energy difference will exist between the energy levels. If an incident electromagnetic radiation is supplied, throughout another magnetic field or a perpendicular frequency (*v${}_{o}$*), there is energy absorption (Δ*E*). The measurement of absorbed energy, recorded as its first derivative, is the EPR signal [42].

Remark that no signal will be detected if the system does not present unpaired electrons, since there will not be resonant absorption of microwave energy. Precisely, molecules with stable nitroxide radicals, such as TEMPO, can be detected. The TEMPO redox sequence involves the oxoammonium cation (TEMPO${}^{+}$) and hydroxylamine (TEMPO-OH) presented in Figure 6, which altogether promote the catalytic oxidation of alcohols [43]. Since the interconversion of TEMPO changes its oxidation state, TEMPO derivatives do not present unpaired electrons; therefore, they will not be detected by the resonance analysis [31].

### 3.2. Chemicals

(2,2,6,6-tetramethylpiperidin-1-yl)oxyl (TEMPO), sodium hypochlorite (NaOCl) (12% Cl), and dimethyl sulfoxide (DMSO) were supplied by Carl Roth. Vanillin (99%) and methyl α-D-glucoside (99%) were provided by Sigma-Aldrich, and silver nitrate (AgNO${}_{3}$) was supplied by Alfa Aesar. Unless otherwise stated, all other reagents were of analytical grade and used without further purification.

The ClO${}_{2}$ solution was prepared by the reaction between sodium chlorite and dilute sulfuric acid (4 N), followed by gas absorption in deionized water [44]. The ClO${}_{2}$ stock solution was stored in complete darkness at 5 °C. The percentage of ClO${}_{2}$ in the solution was determined by the iodometric titration method detailed by Wartiovaara [45], in which the presence of ClO${}_{2}$, chlorite ions, and hypochlorous acid are determined according to their pH dominance. The concentration of the HOCl solution ((HOCl): 0.1 M) was processed by dilution of a reagent-grade NaOCl solution to the pH range 3.5–4.0 with dilute HCl [46].

### 3.3. Technical Parameters for EPR Measurements

#### 3.3.1. Sample Preparation for EPR Analysis

The reactant solutions were prepared in aqueous medium (TEMPO 1 mM, ClO${}_{2}$ 1 mM). The oxidation was controlled for pH (8 and 10) with a boric acid buffer and TEMPO/ClO${}_{2}$ molar ratio (1:1 and 1:2). The reagents for each assay were prepared separately and then combined in an Eppendorf microtube, stirred to homogenize, transferred to a 100 μL glass capillary tube, and then placed inside the spectrometer cavity for the measurements. All trials were performed at room temperature (25 °C).

#### 3.3.2. EPR Measurements Procedure

X-band EPR spectra were recorded with a Bruker EMXmicro spectrometer, equipped with a standard ER4102ST Bruker universal cavity. Typical data acquisition parameters were adjusted, operating at X band: data points, 2500; central field, 360 mT; sweep width, 20 mT; total scans, 3; microwave power, 20.5 mW; modulation frequency, 100 kHz; sweep time, 60 s; and modulation amplitude, 2 G [47]. For each condition evaluated, triplicate samples were made, and EPR measurements were repeated 5 times. Hence, the average intensity of the amplitude signals correspond to the average of 15 scans, which were adjusted with the software Simultispin. For the time-lapse experiments, instant measurement refers to the signal captured after the sweep time, then 3 min and 5 min from the initial data acquisition.

The relative concentration of the nitroxyl radical in each measurement was subjected to the assumption that TEMPO contains only one nitroxyl radical per molecule [48]. Hence, the double integration of each TEMPO spectra after oxidation is calculated as an indicator of the spin density reduction after the oxidation reaction.

### 3.4. UV–Visible Spectroscopy (UV–Vis)

A medium scan speed system is used to register the absorption spectra of TEMPO, ClO${}_{2}$, and TEMPO-ClO${}_{2}$ (at 1:1 and 1:2 molar ratio) to assess the formation of TEMPO${}^{+}$ at different ClO${}_{2}$ concentrations. Absorbance bands were recorded in a UV-1800 SHIMADZU laboratory benchtop spectrophotometer in a wavelength range of 200–450 nm, a scan speed of 1 s (3 times), and a recording range of 0–4 AU (absorbance units).

## 4. Conclusions

A direct conversion of the TEMPO molecule by ClO${}_{2}$-mediated oxidation was proven by EPR and UV–Vis spectroscopy, validating the inner-sphere one-electron oxidation of the organic catalyst. Moreover, the pH 8 and a 1:2 TEMPO/ClO${}_{2}$ ratio kinetic conditions were evaluated in the context of the catalytic oxidation of cellulose, proving that a mildly alkaline medium and higher ClO${}_{2}$ concentration are more suitable for the process in terms of the presence of HOCl as an equivalent oxidant, but this intermediate pH disposition is not entirely advantageous related to the recovery of TEMPO after primary alcohol group oxidation. Furthermore, it is demonstrated that a more defined presence of ClO${}_{2}$ in the process increases the TEMPO${}^{+}$ yield, and its excess after the reaction could be used for another chemical sequence or to be paired with bleaching stages within fibers’ chemical modification, taking into account the selectivity of the oxidant regarding phenolic compounds. Complementary analyses need to be included to clarify the rate of the reaction and the progression of the formation of the nitrosonium ion. This study presents a more specific insight into the oxidation mechanism in the context of cellulose fibers’ oxidation to optimize its use on a bigger scale within a lignocellulosic matrix under different operational conditions.

## Figures and Tables

**Figure 1 molecules-28-06631-f001:**
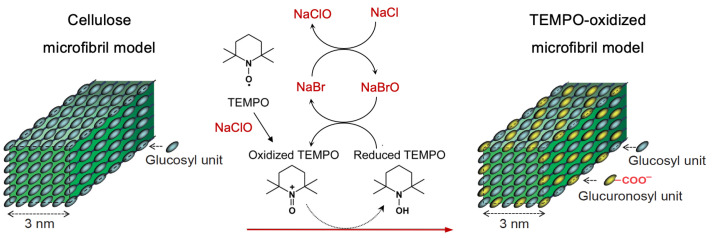
Oxidation of C6 alcohol groups of cellulose by TEMPO/NaBr/NaClO, modified from [2].

**Figure 2 molecules-28-06631-f002:**
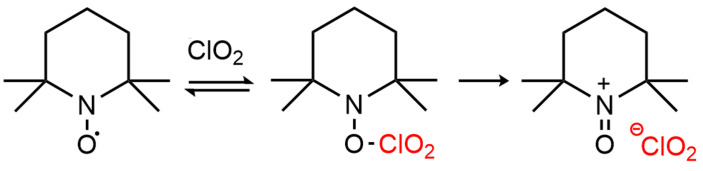
Formation of TEMPO${}^{+}$ via inner-sphere ClO${}_{2}$ one-electron oxidation, adapted from [19].

**Figure 3 molecules-28-06631-f003:**
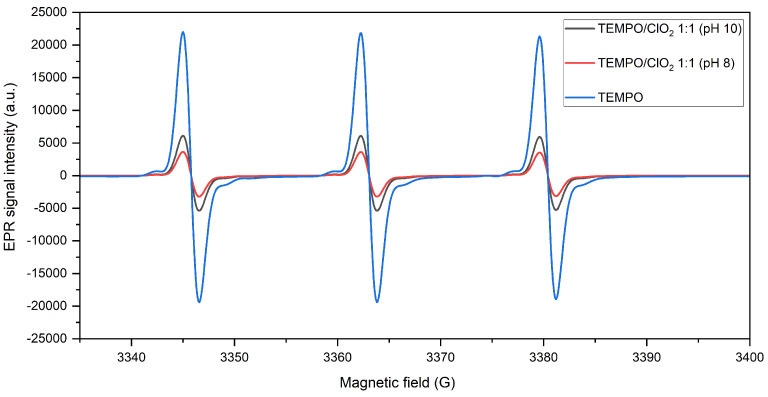
EPR spectra of TEMPO before and after oxidation with ClO${}_{2}$ (1:1 molar ratio) at pH 8 and 10.

**Figure 4 molecules-28-06631-f004:**
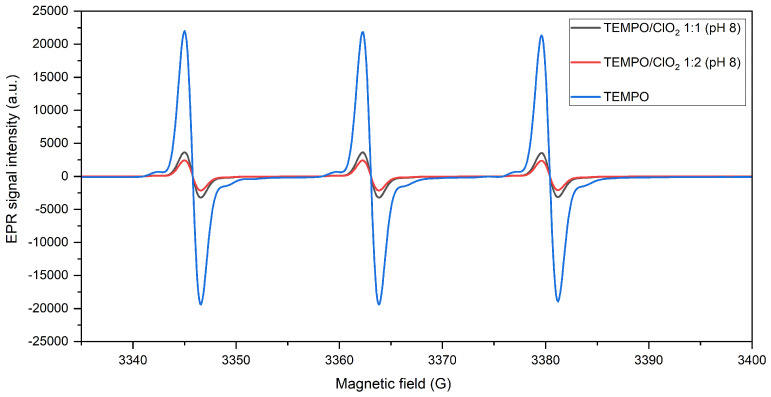
Variation of the TEMPO-ClO${}_{2}$ molar ratio for the ClO${}_{2}$-based TEMPO activation at pH 8.

**Figure 5 molecules-28-06631-f005:**
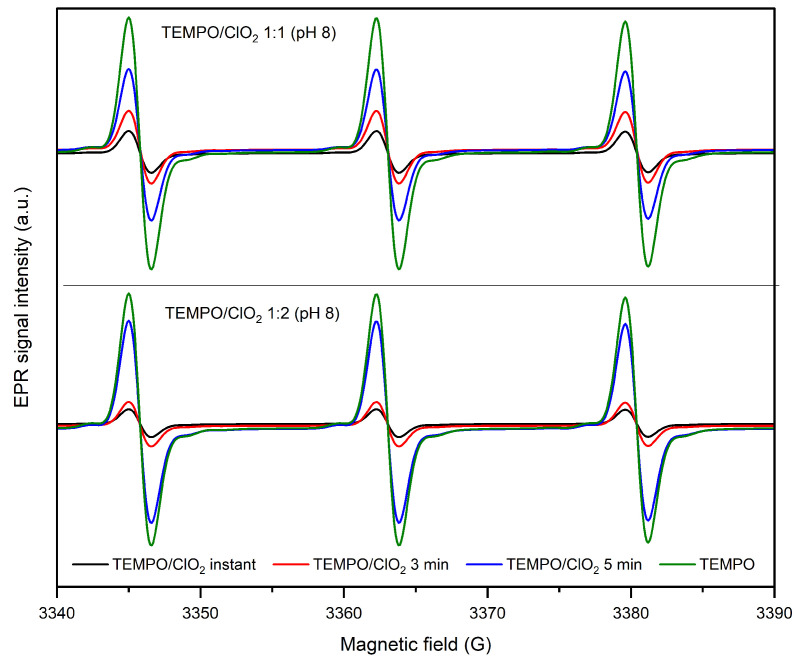
Regeneration of TEMPO after its initial oxidation into TEMPO${}^{+}$; EPR spectra were recorded instantaneously after reaction, then at 3 min and 5 min after reaction.

**Figure 7 molecules-28-06631-f007:**
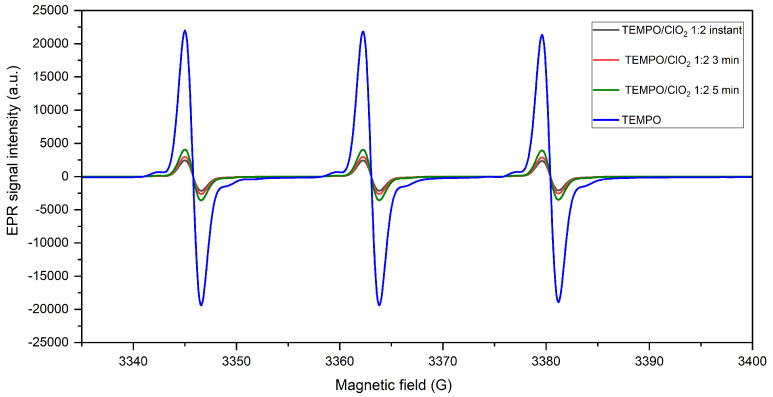
Recovery of TEMPO from TEMPO${}^{+}$ in the absence of chloride (electron donor) at pH 8.

**Figure 8 molecules-28-06631-f008:**
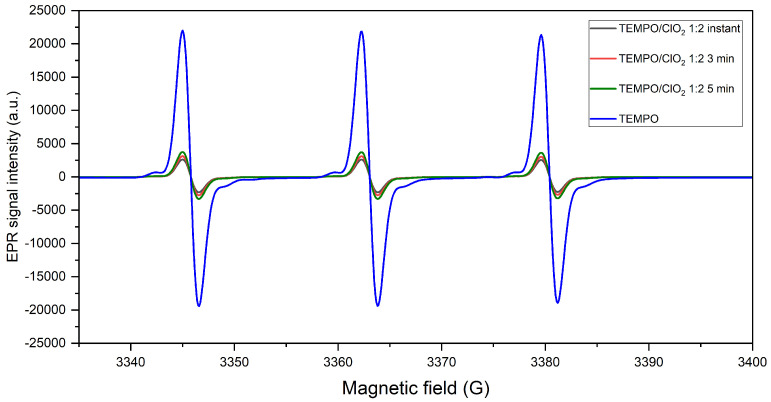
Recovery of TEMPO during oxidation of methyl-α-glucoside by TEMPO-ClO${}_{2}$ at pH 8.

**Figure 9 molecules-28-06631-f009:**
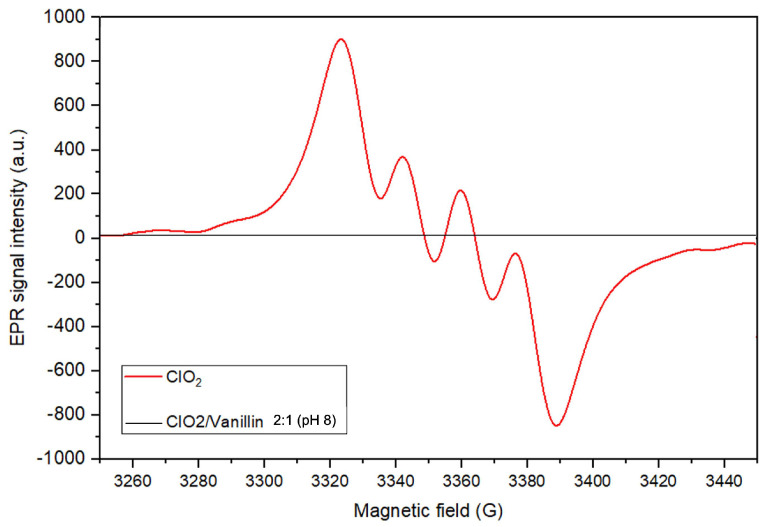
ClO${}_{2}$ EPR signal before and after reacting with vanillin.

**Figure 10 molecules-28-06631-f010:**
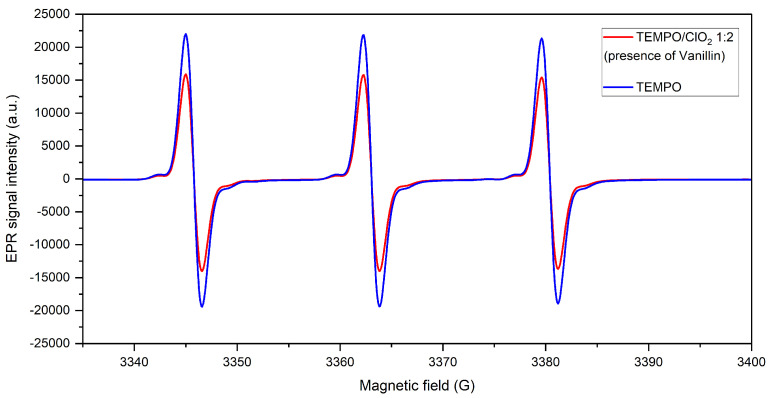
TEMPO-Cl${}_{2}$ reaction in the presence of vanilline.

**Figure 11 molecules-28-06631-f011:**
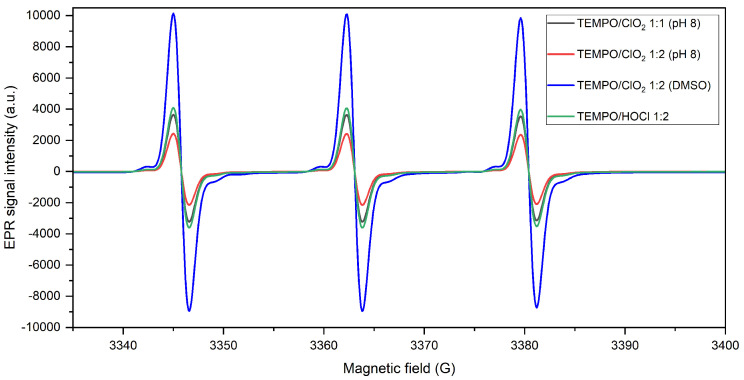
Comparative EPR lines for nitroxyl oxidation in the presence and absence of HOCl.

**Figure 12 molecules-28-06631-f012:**
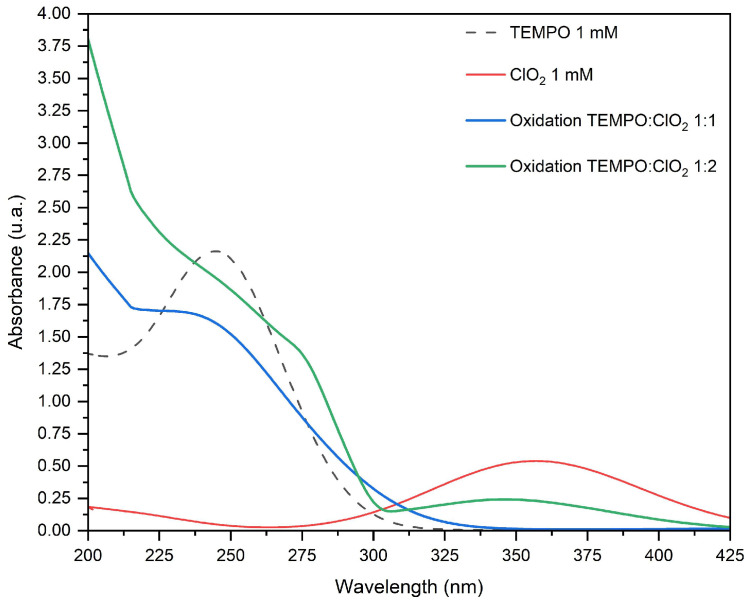
Variation in the UV–Vis spectrum with TEMPO/ClO${}_{2}$ molar ratio.

## Data Availability

The data presented in this study are available in the article.

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
