# Peer review of "ClO_2_-Mediated Oxidation of the TEMPO Radical: Fundamental Considerations of the Catalytic System for the Oxidation of Cellulose Fibers"

_molecules, 2023, doi:10.3390/molecules28186631_

Round 1

Reviewer 1 Report

Please find my comments in the attached pdf file.

Author Response

Thank you very much for taking the time to review this manuscript. We greatly appreciate your perspective and general acknowledgement of the work presented. Please find the detailed responses below and the corresponding revisions/corrections in the document adjoin (they will be finally corrected in the manuscript for the final revision). It is important for us to understand which points were not entirely clear, so they can be improved during the correction process.

Reviewer 2 Report

This manuscript describes the study of the ClO2-mediated TEMPO oxidation in the absence and the presence of cellulose model compound methylglucoside by using EPR and UV-Vis spectroscopy measurements. Some interesting phenomena are observed in this study. However, the results are qualitative and do not lead to quantitative discussion. Therefore, I regret to say that I cannot recommend this paper for publication. Below is a list of inadequacies and points that need to be corrected. I hope that the authors will find it useful for adding experimental data.

1) The title "ClO2-mediated oxidation of the TEMPO radical: Fundamental considerations of the catalytic system" is very vague. The results shown in this manuscript does not provide much detail about the catalytic system. A more specific title that indicates what is clarified in this paper would be better.

2) Figure 2 shows positively charged ClO2 (+ClO2). However, ClO2 is electrically neutral. According to ref 19, ClO2 is not signed +, meaning that the authors have modified the information in the reference paper. For these two reasons, if authors are assuming a positively charged ClO2, need to add a reasonable explanation why authors assume that ClO2 is positively charged.

3) (Line 109) “essay”. Maybe “assay”.

4) (Line 124-125) In this study, authors measured the ESR of aqueous TEMPO solutions. The spectrum presented in Figure 4 represents the pattern of a standard diluted TEMPO solution. Thus, the description "TEMPO displays a standard solid-state spectral pattern" may be miswriting.

5) (Line126) "the signal intensity is diminished." How much of the radical species decreased in this experiment? In ESR measurements, spin quantification allows for quantitative analysis of radical species. Therefore, the ESR results provide data not only for the change in spectral shape but also for a quantitative discussion of the decrease in TEMPO radicals due to the addition of ClO2 and the difference in the extent of the decrease with pH. Considering this point, the description of the experimental results provided here is insufficient.

6) (Line 131 and other parts) Regarding the notation of hypochlorous acid, HClO and HOCl are mixed throughout this paper. Is there some intended distinction in use? If there is a distinction between the two, an explanation should be provided somewhere in the manuscript. If there is no intention, the notation should be consistent throughout the paper.

7) (Line 137-138) Unfortunately, the data are insufficient to make a proof of proportionality. First, one cannot read "TEMPO signal intensity is approximately halved at a 1:2 molar ratio" from the spectrum alone. Second, it is not sufficient to discuss proportionality only for a single data point with a molar ratio of 1:2. To make up for the above points, ESR measurements were performed for at least three different TEMPO/ClO2 ratios, and the relationship between the measured signals and the spin quantification results can be shown to quantitatively discuss the proportionality relationship.

8) (Line 150-153) It would be easier to understand the authors' idea if authors present the reaction equation to show what reaction pathway authors are considering.

If we follow the authors' idea, for example, the following three steps can be considered.

TEMPO radical + ClO2 → TEMPO+ + ClO2-

ClO2- → Cl- + O2

Cl- + TEMPO+ → TEMPO radical + 1/2 Cl2

Alternatively, the following two steps could be considered instead of the third step

2Cl- + TEMPO+ → TEMPO- + Cl2

TEMPO+ + TEMPO- → 2 TEMPO radical

In addition, when considering the above reaction pathway, you should have been able to visually confirm the generation of gases (O2 and Cl2) (formation of bubbles) in the actual experimental operation. Such observations would provide more support for the authors' idea.

9) (Line 156) I think it would be easier to understand if authors clearly describe to which state of TEMPO (TEMPO radical, TEMPO+ etc.) it is restored.

10) (Line 165) “Specifically, disproportionation of TEMPO, as detailed in Figure 10”. The figure is probably Figure 9. Would you please confirm and renumbered.

11) (Line 210-212) The data can be discussed more clearly as competition with vanillin by conducting ESR measurements at different molar ratios to obtain the minimum number of data points required for analysis (at least 3 points) and quantifying the amount of TEMPO radicals by spin quantification.

12) (Line 213-215)“Since ClO2 was not introduced in excess of the phenolic compound, the amount of free oxidant was immediately depleted, leaving a lower quantity to react with TEMPO.” It may be true as an experimental manipulation that no excess was introduced, but the depletion of the free oxidant and the decrease in the reaction volume are speculations based on facts. Fact and speculation should be clearly separated.

13) (Line 254) “Moreover, pH and TEMPO/ClO2 ratio kinetic conditions were evaluated”. The reaction rate is not quantitatively measured in this paper and is not sufficient for the evaluation of kinetic conditions.

14) (Line255) “a mildly alkaline medium and higher ClO2 concentration are more suitable”. The conditions considered in this paper are few. Therefore, I think it is difficult to say SUITABLE. More measurement points need to be added to the analysis to show the validity of the SUITABLE conditions.

15) (Line 263-265) Authors say “This study presents a specific insight of the overall oxidation mechanism”. But, I would hardly say this study provide a specific insight. The measurement conditions are limited, the results are qualitative, and there is no specific discussion of the reaction pathway. Thus, the results seem to be still preliminary data. I think the paper would be more meaningful and of higher quality if there were more experimental results and discussion.

16) Reference list, no. 20, 22, 23, and 32. Perhaps there is a lack of citation information.

Author Response

Thank you very much for taking the time to review this manuscript. Please find the detailed responses below and the corresponding revisions/corrections that will be implemented for the final publication. It is vital for us to improve with your objective comments, and to enrich the quality of my paper.

In the document adjoin, you can find the answers corresponding to the comments made on the paper. Have a very nice day.

Reviewer 3 Report

Before I start giving my opinion on the work "ClO2-mediated oxidation..." by Laura Giraldo Isaza et al., I should state that I am neither an expert in general carbohydrate chemistry and not in the particular field of cellulose treatment, nor am I an experienced EPR spectroscopist. Despite that, I consider the lengthy introduction not necessary for understanding the work, and the section 2.2.1. (fundamentals of EPR) is unneeded in a text directed to an audience probably being aware of this method. The authors could save a lot of space and use it to describe their own work in greater detail!

The authors conducted a extensive series of experiments with components of a system used to perform selective oxidation reactions on cellulose. The reaction solutions have been examined with EPR, and the obtained spectra are brought in relation. This is a kind of quantitative interpretation. As said before, I am not an expert in this method but I remember having read several times that a quantitative evaluation of CW EPR spectra is not a trivial thing. How did you achieve this, "comparability" of absolute signal intensities spectra taken from different samples? Please describe your procedures in very detail. None of the EPR figures has a vertical scale (excluding Fig. 12, why?). Some figures seem to use a spectrum generated with the aqueous TEMPO solution as a kind of reference, but not all (fig. 13) so we cannot draw any conclusion from the signal amplitudes. There are further ambiguities (which must be dealt with) for example:

First, preparation of the solutions (lines 107 to 112). TEMPO is 1 mmol/L, chlorine dioxide is 1 mmol/L. If you mix the two for an "equimolar reaction" (1:1), you end up at 0.5 mmol/L TEMPO and 0.5 mmol/L ClO2. If you mix the two for the 1:2 reaction, you end up at 0.333 mmol/L TEMPO and 0.666 mmol/L ClO2. Even if no reaction occurs, the signals observable for TEMPO will have different intensities. How did you cope with this problem? Is this what line 105 states "amplitude signals...were adjusted"? Please give a detailed description. By the way, signal amplitudes in Figure 5 for the 1:1 solution (black) and the 1:2 solution (blue) incidentally (?) seem to have intensity ratio 3:2, as one would expect from their starting concentration.

Second, reactions and kinetics. The oxidation seems to be fast. So, I assume that the spectra represent a kind of equilibrated state of the system? For the systems described with Figures 6, 8 and 10 this seems not to be the case, there is a development with time, but what about the others? And what is "instantaneously"? Maybe 1 minute, how long does it take to transfer the solution to the spectrometer and then to record the spectrum? Can you add an information on the reaction time for all figures? Is it possible to construct a concentration-time diagram from your measurements? Section 3.1.5. would, for sure, benefit from such a more stringent treatment of the data, as it was hardly possible for me to follow the authors' ideas and interpretation there.

I found it not easy to extract information when regarding the figures. There is not always a glad choice of colours and if you have black-and-white, you are absolutely lost when trying to identify and discriminate the traces. And I suggest drawing lines representing the same species, fx. blank TEMPO, always the same style, in all figures.

I will close EPR issues with a question that maybe sound silly, but as both TEMPO and chlorine dioxide are radicals, isn't it possible to observe both in the same spectrum?

The work is supplemented by UV measurements at the same system. But it is not very clear to me how they are related to the EPR results. Supporting, or complementing, or any new findings? Again, reaction time is missing, and if EPR spectra are time-dependent, the UV curves should behave as well: there is the statement "as the ClO2 curve gradually declines"--with time or with respect to another parameter? More information about the difference spectra (generation and interpretation) would also be helpful.

Many of the facts reported in this study seem to be already known, as I deduce from the respective literature cited. I found it difficult to recognize what is really new and extends knowledge, therefore I suggest that you rewrite the Conclusions section with regard to this issue.

There are a lot of minor issues with language and presentation, I will list only a few (thorough reading is absolutely needed):

* references 20, 22, 23 are incomplete (bibliographic information is missing)

* lines 41-42, what do you mean by "linear dependence in the compound absorption", and is the conversion of TEMPO into the oxoammonium cation a "total oxidation"?

* line 75, "sodium chlorite" correct? I know about the preparation from sodium chlorate + acid, so can you give a reference or a procedure?

* line 124, the TEMPO solution does display a "solid state pattern"?

* line 127, better "regardless of the pH value"

* line 160, is "TEMPO-OH" an established designation for the hydroxylamine? If not, I would consider "TEMPO-H" more appropriate

* line 165, should probably read "figure 9"

* line 166, language: a pH cannot catalyze

* line 176, you state "no noticeable TEMPO restoration", but the according figure 10 is labeled "Recovery" and there is noticeable change in the intensities

* line 178, should probably read "methylglucoside"

* line 201, should probably read "3-methoxy-5-hydroxybenzaldehyde"

* line 203, figure 11 labelling correct (ratio ClO2/vanilline 1:1)--the text says the oxidation "involves two ClO2 (mistyped!) molecules per aromatic unit"?

* line 218, the caption of figure 12 above: should probably read "TEMPO-ClO2"

* line 246, should probably read "possibly"

see "comments and suggestions for authors"

Author Response

Thank you very much for taking the time to review this manuscript. Please find the detailed responses below and the corresponding revisions/corrections that will be implemented for the final publication. It is vital for me to improve with your objective comments, and to enrich the quality of my paper.

Please find adjoin in the document the answers to the comments made, which will be taken into consideration for the improvement of the paper. Have a very nice day

Round 2

Reviewer 1 Report

I think the manuscript has been improved and can be published now.

the language is ok

Reviewer 2 Report

Although the quantitative evaluation of ESR measurement results cannot be said to be sufficient, I think that the authors have done their best to revise the results by adding the minimum amount of information. The readability has improved compared to the previous manuscript, and I think it is worth publishing.